# Surveillance of Infections and Antibiotic Use in 21 Nursing Home Wards during the COVID-19 Pandemic: A Systematic Assessment

**DOI:** 10.3390/ijerph21030358

**Published:** 2024-03-18

**Authors:** Ida Hellum Sandbekken, Åsmund Hermansen, Ellen Karine Grov, Inger Utne, Borghild Løyland

**Affiliations:** 1Department of Nursing and Health Promotion, Faculty of Health Sciences, Oslo Metropolitan University, 0130 Oslo, Norway; idahan@oslomet.no (I.H.S.); ellgro@oslomet.no (E.K.G.); inger@oslomet.no (I.U.); 2Department of Social Work, Child Welfare and Social Policy, Oslo Metropolitan University, 0130 Oslo, Norway; asmuhe@oslomet.no

**Keywords:** infections, healthcare-associated infections, antibiotic use, nursing homes, COVID-19

## Abstract

Residents in nursing homes are fragile and at high risk of serious illness or death from healthcare-associated infections. The COVID-19 pandemic posed a significant risk of suffering and mortality for residents of nursing homes. Surveillance of infections is essential for infection prevention and is missing in many countries. The aim of this study is to explore infection rates and antibiotic use in nursing homes during the COVID-19 pandemic. Data collection was conducted from February to September 2021. Each week, healthcare workers at 21 nursing home wards answered a questionnaire on infections, antibiotic use, deaths, and hospital admissions related to infections. A total of 495 infections were reported, and 97.6% were treated with antibiotics. The total infection rate was 5.37 per 1000 bed days, and there were reported 53 hospital admissions and 11 deaths related to or caused by infections. The infection rate and high use of antibiotics found in this study indicated that it is difficult to treat infections in residents in nursing homes and make it difficult to achieve the global goal of reducing infections and antibiotic resistance rates. This emphasizes the need for stricter infection control programs to reduce antibiotic use and patient suffering.

## 1. Introduction

Surveillance of healthcare-associated infections (HAIs) is an essential element of infection control programs [1] and is important in order to quickly detect and initiate local interventions against infections [2]. Few countries have surveillance of infections in nursing homes, and there are no data on how many residents acquire HAIs throughout the year [2]. According to a European point prevalence study conducted in 2016 and 2017, 3.8% of residents in nursing homes were found to have at least one HAI. However, there were significant variations observed, ranging from 0.9% to 8.5% across the different countries [3]. Point prevalence to monitor infections may provide some insight, but it does not capture the whole picture. This can be addressed using continuous surveillance over a period [4].

Worldwide, the population of older people is steadily increasing [4,5], and this is one of the most vulnerable groups to experience serious consequences of an infection. In Norway, the most fragile older people are prioritized for nursing homes [6]. Most of the residents in nursing homes are older people with comorbidities or reduced functional status and immune systems and have an increased risk of acquiring an infection [7,8,9,10], and around 84% of the residents suffer from dementia [11]. Nursing home residents retrieving infections experience longer hospital stays, suffering, and death, and many of these infections could be avoided [12,13]. The coronavirus disease (COVID-19) posed a significant risk for residents in nursing homes, and an important fraction of the total number of deaths emerged from nursing homes [14]. The pandemic showed that there is a need for surveillance of infections in nursing homes and to increase the focus on infection prevention for this vulnerable group.

Nursing home residents who acquire an infection need antibiotic treatment more often than older people living at home [15] and have atypical symptoms, problems with communicating symptoms, and cognitive impairments [7,10]. Therefore, knowing when and what types of antibiotics to prescribe in nursing homes may be difficult, and a significant proportion of these prescriptions may be unwarranted [7]. Unnecessary antibiotic use contributes to antibiotic resistance problems, can cause side effects for the residents, and may increase costs [7,16,17]. Surveillance of antibiotic use in nursing homes provides important insight into antibiotic trends, and deviations from these trends indicate a need to implement infection prevention interventions.

Studies on antibiotic use and resistance have mainly focused on hospitals [10,17]. The lack of knowledge concerning antibiotic use [10,18] and infections in long-term care facilities shows that there is an increased need for studies on this health service. COVID-19 constituted a special situation for nursing homes, with a high focus on infection prevention, making it interesting to investigate all infections in nursing homes during the pandemic. To meet the international goal [1,19] to reduce infections and antibiotic resistance rates, we need to conduct surveillance in nursing homes. Therefore, the aim of this study was to examine infections and antibiotic use in 21 nursing home wards over six months during the pandemic.

## 2. Materials and Methods

### 2.1. Design

This study is a quantitative prospective study in which a report questionnaire was used to gather data on infections and antibiotic use. It is part of a larger research project aimed at investigating hand hygiene practices, as well as infection and antibiotic use in nursing homes. The overarching objective of the research project is to explore whether interventions targeting behavior change can enhance hand hygiene adherence and subsequently reduce infections [20,21].

### 2.2. Sample and Recruitment

In December 2020, 17 nursing homes in one municipality in Norway were sent invitation letters to participate in a study on infection prevention from January 2021 to July 2022. A total of 21 nursing home wards from nine nursing homes agreed to participate in the research project. The wards sent weekly reports of infections and antibiotic use from February to September 2021. This occurred during the COVID-19 pandemic, and nursing homes were closed for visits from relatives and others for periods. The quality manager, ward leader, or institution leader signed an agreement for each ward and sent it to the first author before the study began.

### 2.3. Questionnaire and Data Collection

The leaders themselves or other healthcare workers, such as nurse assistants, nurses, or doctors, were assigned the task of sending in the reports once a week. Each ward was given an individual code number for identification. Only the first author had access to the codes. Information about the code numbers was given by phone, and the identification documents were locked in a safe at Oslo Metropolitan University. If a ward did not send a report, they received a reminder by e-mail the following week.

Data were gathered using Nettskjema, a tool for secure online data collection that was developed to comply with Norwegian privacy requirements. Nettskjema sends the information automatically to TSD (Tjeneste for sensitive data). The TSD service is designed for storing and post-processing sensitive data in compliance with the “Personal Data Act” and “Health Research Act”. TSD is owned by the University of Oslo, and it is operated and developed by the TSD service group at the University of Oslo, IT Department (USIT).

The survey included questions on how many residents had infections, their sex, and age (≤84 or ≥85), when the infection arose, type of infection, whether the residents had more than one infection, use of antibiotics, type of antibiotics, predicted length of treatment and start date, hospital admissions, and infection-related deaths (the full version of the questionnaire is available in the Appendix A). The questionnaire was pilot-tested in one nursing home ward before the study began, and both the length of the questionnaire and the most used types of antibiotics were modified based on feedback from the ward. All wards received a written manual on how to complete the report, with details for each question. The first author also had a phone conversation with each ward, during which time they reviewed the manual. They also received the contact information for the first author, along with information that they could reach out if they were unsure about the report questionnaire or had any other questions. All types of infections found in the residents were supposed to be reported. In nursing homes, doctors diagnose the infection. This diagnosis is often based on information about the patient’s symptoms that the nurses gather. It is also the doctor who prescribes the antibiotics. In this study, no microbial cultures were sent in or analyzed. In April, one question with the date of registration was added, which made it possible to post-register. To minimize the questionnaire length, a once-only Nettskjema questionnaire was sent out in June 2021 that included questions about ward type (short- or long-term), total number of beds and residents, sex, and age of residents (≤84 or ≥85), full-time staff positions, and full-time nursing positions.

The first author maintained close contact with the wards during the 18 months of data collection. All wards received a reminder via email if they did not send in the weekly report. In addition, if there was no response to the email and reports continued to be missing, the wards would receive a phone call. This happened on multiple occasions due to staff changes, maternity leaves, and reorganization of the nursing homes. Some wards also received a visit from the researchers to increase motivation to continue sending in the reports.

### 2.4. Data Analysis

Data were analyzed using STATA 16th edition for Windows. Variables written in the free text were categorized. Descriptive and linear regression analyses were performed. Infection rates were reported as infections per 1000 bed days. To calculate bed days, the total days participating in the study were multiplied by occupied beds from each ward. Total infections were divided by total bed days and multiplied by 1000. Estimated infections (Figure 1) were calculated by adding the mean weekly infection rate from each ward to the weeks of missing reporting. Significant variables in bivariate linear regression analyses were included in a standard multiple linear regression analysis, where all independent variables were entered into the equation simultaneously. The number of infections each week was used as the dependent variable. *p* < 0.05 was considered statistically significant.

### 2.5. Ethical Considerations

The Regional Committee for Medical and Health Research Ethics (REC), Norway (Ref. 196911 and 226694/REC South-East) and the Norwegian Center for Research Data (Ref. 118936) reviewed the project.

## 3. Results

### 3.1. Demographic Data

A total of 21 nursing home wards agreed to complete the weekly questionnaire. These 21 wards consisted of 18 long-term wards and 3 short-term wards. They were all from the same metropolitan area. Sociodemographic data, with total, mean, and range for each ward, are shown in Table 1.

### 3.2. Infections

A total of 495 infections were reported during the six months. There were a total of 53 hospital admissions and 11 deaths related to or caused by infections. The rates of infections remained relatively stable, with peaks in weeks 14 and 31 and the lowest rates in weeks 12 and 32 (Figure 1). Weeks are reported on the x-axis, with week numbers for 2021. The number of infections is reported on the y-axis. Estimated infections are used to estimate the number of infections reported if all wards send in reports each week.

Forty-seven percent of the infections reported were urinary tract infections, followed by respiratory tract infections and skin tissue infections (Table 2). The category named “other” is a summed category combining all other infections, such as ear and eye infections, tuberculosis, bone infections, and endocarditis. The total infection rate was 5.37 per 1000 bed days. Respiratory tract, skin tissue, and gastrointestinal infections were distributed equally between genders. Urinary tract infections were overrepresented in women (64.9%) and sepsis in men (71.4%). 

### 3.3. Antibiotic Use

A total of 97.6% of the infections were treated with antibiotics. Of the respiratory tract infections, only one did not get antibiotic treatment; for urinary tract infections, this number was two; for skin and tissue infection, it was six (6.6%); and for infections with unknown focus, only one person with infection did not get antibiotic treatment. The most commonly prescribed group of antibiotics was J01C beta-lactam antibacterial penicillin, followed by J01E trimethoprim–sulfonamide (Table 3). Altogether, 41 different types of antibiotics were prescribed. Prescriptions were oral administration in 87.3% of cases and intravenous in 9.1%.

### 3.4. Regression Analyses

Standard multiple linear regression analyses of the sociodemographic variables and the total number of infections from each ward found some small but significant relationships (Table 4). The largest significant finding was ward type, where being a resident in a short-term ward had a stronger association with infection than being a resident in a long-term ward.

## 4. Discussion

To the best of our knowledge, this is the first study to examine infection rates and antibiotic usage in nursing homes over six months during the COVID-19 pandemic. Urinary tract infections were the most common type, and amoxicillin and trimethoprim–sulfamethoxazole were the most frequently prescribed antibiotics. Residents in short-term wards had an increased risk of infection. Only one respiratory tract infection was not treated with antibiotics, indicating that there were few residents in nursing homes that were reported with COVID-19 during the data collection period. It is also possible that some infections were mistakenly treated as bacterial infections and were inappropriately prescribed antibiotic treatment when they were actually caused by viruses. COVID-19 was a relatively new disease in 2021, and some doctors may have had difficulty differentiating it from other bacterial infections. However, in 2021, COVID-19 rapid antigen tests were available for both health institutions and the general population, so a test would likely have been performed before starting antibiotic treatment. Another potential explanation for the low rates of possible COVID-19 infections could be that many of the residents who contracted COVID-19 were transferred to designated COVID-19 wards. Furthermore, the vaccination of residents in nursing homes was completed in February 2021. Nursing home residents were prioritized for vaccination and were the first to receive it in Norway due to the vulnerability of the residents and the high number of deaths in the early stages of the pandemic [14].

A surprising finding was that antibiotics were prescribed for almost all infections, both in long- and short-term wards. Although antibiotics are a frequently prescribed drug in long-term care facilities [18], research indicates that not all infections need to be treated with antibiotics [7,16]. A percentage of 97.6% is higher than earlier studies from Norwegian nursing homes that indicated a range of 76–79% [16] and 94% [22]. A possible explanation for this might be that healthcare workers are prone to only remember or recognize and report infections that require antibiotics, as they are easier to detect by reviewing the residents’ medication lists. In addition, older people often have cognitive impairments, problems with communicating symptoms, and atypical symptoms [7,10], and this may have resulted in lower reports of infections than there really were. Residents in nursing homes are at high risk of having asymptomatic bacteriuria, which typically does not need antibiotic treatment [23]. The use of unwarranted antibiotics can increase costs, side effects, and antibiotic resistance [7,10,16,17,24]. On the other hand, residents in nursing homes are some of the most fragile and vulnerable populations, and they have an increased chance of hospitalization and death as a result of infection [12,13]. Preventing HAIs in nursing homes is crucial not only for reducing suffering and death caused by infections but also for restraining the high use of antibiotics that can contribute to higher antibiotic resistance rates.

Amoxicillin is a broad-spectrum antibiotic [25], and we found it was the most frequently used, together with pivmecillinam. Amoxicillin was most often used for respiratory tract infections, even though phenoxymethylpenicillin is the recommended antibiotic for most upper and lower respiratory tract infections, and amoxicillin is only recommended when a resident has chronic obstructive pulmonary disease (COPD) [26]. This may indicate that many residents in the nursing homes had COPD diagnoses. However, it could also indicate that doctors at nursing homes often use broad-spectrum antibiotics because of more complex and multimorbid patient conditions. Following the recommendations may be difficult in nursing homes because residents often have several chronic diseases, atypical symptoms, communication difficulties, and/or cognitive impairments [7,10]. In this study, we do not have detailed information about each resident and infection and could not determine whether antibiotic use was inappropriate. This is a limitation of this study. At the same time, the types of antibiotics used to treat the different infections largely adhere to the antimicrobial stewardship rules applied in Norway [26]. This may indicate that nursing homes are largely following these recommendations.

Another finding in this study was an infection rate of 5.37 per 1000 bed days. This finding is in the middle but tends toward high compared with findings from other studies. Previous international studies from nursing homes found infection rates ranging from 2.7 to 7.0 per 1000 bed days [4,10,12,27,28]. The present study found that infection rates varied substantially between the different weeks; these findings are interesting in light of the ongoing pandemic, with a high focus on infection prevention. However, the peaks and lows found are not in line with the peaks and lows of the pandemic. Even so, the pandemic can explain some of the findings. Infections normally vary depending on the season, with higher numbers anticipated in winter compared to spring and summer [29]. Nevertheless, during the pandemic, strict regulations disrupted this pattern. At the beginning of 2021, no visitors were allowed in nursing home wards, and healthcare workers with potential COVID-19 symptoms had to isolate. This resulted in lower numbers of hospital admissions due to respiratory tract infections than usual [30]. When these strict regulations were softened, we observed higher infection rates [30]. As demonstrated in this study, the highest infection rates were during the summer.

## 5. Strengths and Limitations

The inclusion of 21 nursing home wards is considered a strength of the study, together with the fact that no ward withdrew from the study during the data collection period. Additionally, to the best of our knowledge, no studies from Norway have taken a prospective perspective on infections in nursing homes. One limitation was that there were several missing weeks, and one ward did not send a report for 13 of the 31 weeks of data collection. Additionally, only collecting data over six months can cause us to lose important data on seasonal variations and additional COVID-19 outbreaks. Only including February as a winter month may have influenced the results and underestimated the infection rates, especially for respiratory tract infections. Self-report questionnaires may result in reporting that differs from actual events. In this study, it is possible that some infections have not been reported due to a lack of recognized infections because of atypical symptoms in older people. It is a limitation that we did not follow the specific residents in the nursing homes. This has resulted in a lack of knowledge about the residents’ underlying diseases, medical equipment, or the outcome of an infection. We included very few sociodemographic questions in the weekly reporting, and the questionnaire with sociodemographic data was only sent out once. The frequent changes in sociodemographic data within the short-term wards over the period limited the analyses. The first author’s continued contact with the wards and the short weekly questionnaire were factors of importance for us and possibly a reason for experiencing no dropouts.

## 6. Conclusions

This prospective study explored infection rates and antibiotic usage in 21 nursing home wards over six months. During the study period, 495 infections were reported. Even though the current research was conducted during the COVID-19 pandemic, urinary tract infections were the most common, followed by respiratory tract infections. Antibiotic usage is generally low in Norway, but this study found that 97% of all infections were treated with antibiotics. Such results indicate that the extensive use of antibiotics in nursing homes is still a problem.

The research has shown a moderate to high incidence rate of infections in nursing homes and the frequent use of antibiotics for treatment. There is an international goal to have ongoing surveillance of infections as part of an infection prevention program. Surveillance of infections enables rapid responses when infection rates deviate from the mean and reduces the spread of infections. Many HAIs are preventable, and it is necessary to prevent infections in order to reduce antibiotic use in the fight against antibiotic resistance. Future studies should focus on infection prevention in nursing homes. By focusing on this area, the research seeks to address a critical issue in nursing homes, which causes suffering, decreased quality of life, and even death among vulnerable residents. Improved infection prevention can enhance the well-being and life expectancy of many older people.

## Figures and Tables

**Figure 1 ijerph-21-00358-f001:**
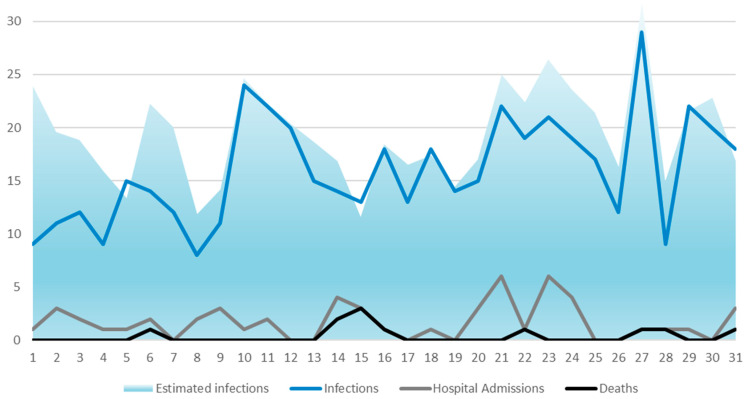
Weekly reports of numbers of infections, estimated infections, hospital admissions, and deaths.

**Table 1 ijerph-21-00358-t001:** Sociodemographic data, with total, mean, and range for each ward.

	Total	Mean	Range
Beds	611	27	18–32
Beds occupied	585	95.7%	83.3–100%
Female residents	407	69.9%	43.5–88.2%
Age 85 years or above	341	58.2%	8.7–100%
Weeks of reporting infections	31	24	18–31
Full-time positions	501.3	22.3	13–35
Full-time nurses’ position	153.6	3.7	1.7–5.3
Beds per healthcare worker	1.22	1.25	0.91–1.88

**Table 2 ijerph-21-00358-t002:** Distribution of total infections by type of infection.

Infections	% (n)	Infections per 1000 Bed Days
Urinary tract infections	46.9% (232)	2.52
Respiratory tract infections	19.2% (95)	1.03
Skin tissue infections	18.4% (91)	0.99
Unknown focus	4.4% (22)	0.24
Sepsis	1.4% (7)	0.08
Gastrointestinal infections	1.2% (6)	0.07
Other	8.5% (42)	0.46
Total	100% (495)	5.37

**Table 3 ijerph-21-00358-t003:** Number of prescribed antibiotics distributed by type of antibiotic and type of bacterial infection.

Antibiotics	Urinary Tract Infections	Respiratory Tract Infections	Skin Tissue Infections	Sepsis	Gastrointestinal Infections	Unknown Focus	Other	Total	%
Amoxicillin	28	42	4	0	0	6	1	81	16.8
Trimethoprim–sulfamethoxazole	63	1	13	2	1	1	0	81	16.8
Pivmecillinam	66	0	0	0	0	0	0	66	13.7
Dicloxacillin	3	1	39	0	0	2	2	47	9.8
Ciprofloxacin	16	8	4	0	0	3	3	34	7.1
Phenoxymethylpenicillin	3	15	9	1	0	1	4	33	6.9
Trimethoprim	25	0	0	0	0	0	0	25	5.2
Cefotaxime	14	8	1	0	0	2	0	25	5.2
Combination/multiple types	5	8	1	0	1	0	1	16	3.3
Other	7	10	12	4	4	5	31	73	15.2
Total	230	93	83	7	6	20	42	481	

**Table 4 ijerph-21-00358-t004:** Standard linear regression analyses of sociodemographic variables with the number of infections as the dependent variable.

Variable	Bivariate Analyses			Multivariate Analysis		
	B	95% CI	*p*-Value	B	95% CI	*p*-Value
Constant				−3.71	−5.78–−1.65	<0.001
Short-term ward	1.75	1.52–1.97	<0.001	1.57	1.03–2.11	<0.001
Percentage of occupied beds	−0.04	−0.06–−0.03	<0.001	0.05	0.02–0.07	<0.001
Percentage over 85 years old	−0.02	−0.02–−0.01	<0.001	0.01	0.01–0.02	<0.001
Percentage being women	−0.04	−0.05–−0.03	<0.001	−0.02	−0.03–−0.01	0.003
Hospital admissions	0.51	0.26–0.75	<0.001	0.41	0.19–0.62	<0.001
Percentage nurse positions	0.25	0.22–0.29	<0.001	0.11	0.05–0.18	0.001

*p* < 0.05 was considered statistically significant.

## Data Availability

The data that support the findings of this study are available upon request from the corresponding author, BL. The data are not publicly available due to them containing information that could compromise research participants’ privacy.

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
