# Peer review of "Surveillance of Infections and Antibiotic Use in 21 Nursing Home Wards during the COVID-19 Pandemic: A Systematic Assessment"

_ijerph, 2024, doi:10.3390/ijerph21030358_

Round 1

Reviewer 1 Report

Comments and Suggestions for Authors

Comment for the authors,

Was there direct supervision by the authors during the study to ensure how to fill out the questionnaire?

It is not clear in the text on what basis the infection was diagnosed in the study population. Has a specialist doctor diagnosed it? The criteria for infection of residents should be determined.

Has microbial culture been sent from the studies?

Line 92-3: "All types of infections, bacterial, viral, parasitic, fungal, and unknown, were supposed to be reported."  Why is it not reported by the type of infection (bacterial, viral, parasitic, fungal) in the results section?

Antibiotics are prescribed by whom for antibiotic patients in the nursing homes? To be determined.

Table 2: Is the source of infection clear? Bacteria, fungi or...

In Table 3. Bacterial infection should be mentioned in the title.

Were 7 patients who were diagnosed with sepsis admitted to the hospital? If they were hospitalized, were they treated with the antibiotics listed in Table 3 or with meropenem or vancomycin and...

It seems that the diagnosis of infection was not based on paraclinical findings. Please explain in the discussion section.

Is it possible that antibiotics have been wrongly prescribed to residents in respiratory infections caused by viruses or not? And in fact, bacterial infection is reported instead of viral infection. Please explain in the results or discussion section.

Reviewer 2 Report

Comments and Suggestions for Authors

The study reports  the results of a multicentre surveillance for healthcare acquired infections and antibiotic use in 21 nursing homes over 6 months in Norway; since most studies regarding long term care are prevalence studies, the  fact that the present paper reports infection rates per 1000 bed days over 6 months is surely a positive issue.

Several limits, besides those indicated by the authors, need however to be pointed out. The winter months were not included, so it is possible that the real rate of respiratory tract infections is underestimated.  Since the patients admitted to nursing homes are highly heterogeneous as regards their clinical status and care needs the fact that only sociodemographic variables were evaluated and no indication is given regarding risk factors linked to  patients condition, such as underlying diseases and exposure to devices - urinary or vascular catheters, ventilation, enteral or parenteral nutrition...- is a strong limit to the appraisal of the reported results and comparison to other studies.The infections  observed are only reported in Table 2 as a percentage of the overall infection rate and no data about the incidence of each kind of infection is given.  Antibiotic use is reported as percentage of infections treated instead of treated patients and no data about overall prescription is given, eg the number of  antibiotic days/ 1000 bed days. It should positively noticed however that the classes of the prescribed antibiotics reflect the antimicrobial stewardship rules applied in Norway which are more strict than those followed in most European countries.

The symbols used to identify the hospital admissions and deaths in Figure 1 may be confusing.

The questionnaire and the data collection form could be useful attached as an appendix to the article.

Round 2

Reviewer 1 Report

Comments and Suggestions for Authors

Thanks to the respected authors for making corrections.

No comment.

Reviewer 2 Report

Comments and Suggestions for Authors

The manuscript has been duly revised according to the comments by the reviewers and may be published in the present form